# Effects of Different Galacto-Oligosaccharide Supplementation on Growth Performance, Immune Function, Serum Nutrients, and Appetite-Related Hormones in Holstein Calves

**DOI:** 10.3390/ani13213366

**Published:** 2023-10-30

**Authors:** Xin Yu, Fengtao Ma, Haonan Dai, Junhao Liu, Nesrein M. Hashem, Peng Sun

**Affiliations:** 1State Key Laboratory of Animal Nutrition and Feeding, Institute of Animal Science, Chinese Academy of Agricultural Sciences, Beijing 100193, China; 2Department of Animal and Fish Production, Faculty of Agriculture, Alexandria University, Alexandria 21545, Egypt; nesreen.hashem@alexu.edu.eg

**Keywords:** galacto-oligosaccharides, heifers, growth performance, immunity, microelement, appetite-related hormones

## Abstract

**Simple Summary:**

This study aimed to investigate galacto-oligosaccharide (GOS) doses and their effects on growth, immune status, nutrition, and appetite in Holstein heifer calves. Twenty-eight newborn healthy Holstein heifer calves were randomly divided into four groups, receiving milk supplemented with 0, 2.5, 5, and 10 g/(d·head) GOS for 28 days. We found that the calves receiving 5 g/(d·head) of GOS experienced better growth performance, increased feed intake, enhanced immune function, and nutrient balance. This suggests that GOS supplementation has a potential application in calf rearing.

**Abstract:**

Our previous study showed that early supplementation with 10 g/(d·head) of galacto-oligosaccharides (GOS) in newborn Holstein dairy calves reduced the incidence of diarrhea and improved growth performance and mineral absorption. Since the dose of 10 g/(d·head) was the lowest by dose screening in our previous study, the present study was designed to investigate whether a lower amount of GOS has similar effects on growth performance, immune function, serum nutrients in newborn Holstein heifer calves, and to further investigate its effect on appetite-related hormones. Twenty-eight healthy newborn (1 day of age) Holstein heifers with similar average body weight (41.18 ± 1.90 kg) were randomly divided into four groups (*n* = 7): the control group (CON group), which received heated raw milk, and three experimental groups, which received heated raw milk supplemented with 2.5 (GOS2.5 group), 5 (GOS5 group), and 10 g/(d·head) (GOS10 group) GOS. All heifer calves were fed the same starter for 28 d. Supplementation with GOS linearly increased the final body weight, average daily gain, and feed efficiency in heifer calves (*p* < 0.01). Compared with the control group, the average daily gain and feed efficiency of heifer calves were significantly higher in the GOS5 and GOS10 groups than in the control group (*p* < 0.05). Furthermore, supplementation with GOS quadratically enhanced the starter and total average daily feed intake of the heifers (*p* < 0.01), especially in the GOS2.5 and GOS5 groups, (*p* < 0.05 vs. CON). The serum concentration of immunoglobulin A was linearly increased by GOS supplementation (*p* < 0.05), and the levels in the GOS5 and GOS10 groups were significantly higher than those in the CON group. Meanwhile, GOS linearly decreased serum interleukin-1β and interleukin-6 concentrations (*p* < 0.05). The serum concentration of triglycerides was also linearly decreased (*p* < 0.05), whereas total protein and blood urea nitrogen were linearly increased (*p* < 0.05). Supplementation with GOS linearly decreased the serum concentration of leptin (*p* < 0.05) but increased cholecystokinin and glucagon-like peptide-1 (*p* < 0.05). Increasing doses of GOS linearly improved serum calcium and copper concentrations (*p* < 0.01) and quadratically enhanced the concentration of magnesium, which peaked in the GOS5 group (*p* < 0.05). In conclusion, GOS supplementation reduced the incidence of diarrhea and improved the growth performance and immune function of Holstein heifer calves.

## 1. Introduction

The health of dairy calves is of great importance for the future production of dairy cows. Diarrhea is the most common disease in calves and occurs frequently, especially during early life [1]. Given the global efforts geared toward antimicrobial-free farming for achieving a more eco-friendly and sustainable dairy industry, there is a growing need to discover safe and natural alternatives to antimicrobials for improving calf growth and lowering calf morbidity and mortality [2,3].

The International Scientific Association for Probiotics and Prebiotics (ISAPP) defines prebiotics as “fermentation substrates that are selectively utilized by host microorganisms for health benefits”. Several prebiotics have been identified, including oligosaccharides, resistant starch, and non-starch polysaccharides [4]. Prebiotics play an important role in feeding micro-ecological preparations and are potential biological alternatives to antibiotics, as they can prevent intestinal infections by selectively promoting the growth of beneficial intestinal bacteria. Hence, they enable these beneficial bacteria to colonize the ecological niche of the host intestine, competitively rejecting pathogens [5]. Furthermore, prebiotics can directly participate in and regulate the host immune response through immunomodulation and immune stimulation, reducing chronic inflammation and allergic disease symptoms [6].

Galacto-oligosaccharides (GOS) are a natural group of oligosaccharides with the molecular structural formula Gal-(Gal)_p_-Glc/Gal (*p* = 0–6). They consist of a terminal glucoside and 1 to 7 galactoside residues, which are usually produced by β-galactosidase-mediated enzymatic cleavage using lactose as a donor and acceptor molecule [7]. The monomers that make up GOS are linked by β-glycosidic bonds. Hence, they exhibit strong acid resistance and thermal stability. GOS can resist decomposition by saliva, gastric acid, and digestive enzymes and can, therefore, reach the hindgut in large amounts. The intestinal microbiota can thus effectively decompose and utilize GOS. This improves the structure of the intestinal microbiota, as well as the immune and antioxidant function, and also regulates glycolipid metabolism [8,9].

GOS has anti-adhesive properties. An in vitro cellular assay in lipopolysaccharide (LPS)-induced RAW264.7 macrophages showed that GOS inhibits nitric oxide (NO) production and reduces the release of pro-inflammatory factors [10]. Quigley et al. [11] found that GOS substituted for 1% dry matter milk powder increased feed efficiency and nutrient utilization to improve the growth performance of calves. Short et al. [12] found that GOS increased the apparent efficiency of absorption of IgG in calves by 20% compared to the control group. Mineral deficiencies in young animals may lead to growth retardation and deficiency, but supplementation with GOS can promote the absorption of minerals such as calcium, iron, zinc, and magnesium [13,14,15,16,17,18,19,20]. Chang et al. [21] found that GOS significantly increased the serum calcium concentrations in calves. In addition, GOS can be fermented by intestinal microorganisms to produce short-chain fatty acids (SCFAs), which act as important signaling molecules to regulate lipid metabolism via the gut–liver axis [21,22]. Castro et al. [23] found that GOS supplementation increased the relative abundance of lactic acid bacteria in the colons of calves and improved the SCFA profile during the first 2 weeks.

Our previous study demonstrated that early feeding with 10 g/(d·head) GOS can significantly enhance growth performance, reduce the incidence of diarrhea, and improve lipid metabolism and mineral absorption in newborn calves [21]. Since this dose was the lowest in our previous research of dose screening [24], we assumed that the effective dose might be even lower, and different amounts of GOS, i.e., 2.5, 5, and 10 g/(d·head), were set in the present experiment. Therefore, we hypothesized that lower amounts of GOS might exhibit similar beneficial effects on growth performance, diarrhea incidence, and immune function in calves through regulating nutrient absorption and appetite hormone. Here, the purpose of this study was to detect the effects of different doses of GOS supplementation on growth performance, diarrhea incidence, immune function, serum biochemical indicators, mineral levels, and appetite-related hormones in dairy calves.

## 2. Materials and Methods

The experimental protocol of this study was approved by the Experimental Animal Ethics Committee of the Institute of Animal Science, Chinese Academy of Agricultural Sciences (IAS, CAAS, Beijing, China) (approval number IAS2020-102). All the procedures involved were in accordance with the standards stated in the “Guidelines for the Management and Use of Laboratory Animals” of the IAS, CAAS.

### 2.1. Experimental Animals and Design

Twenty-eight newborn (1 day of age) Holstein heifer calves with similar birth weights (41.18 ± 1.90 kg) were randomly allocated into four groups (*n* = 7) using a random number generator. The heifers in the control group (CON group) were fed with heated raw milk, whereas the heifers in the experimental groups received heated raw milk supplemented with different amounts of GOS, i.e., 2.5, 5, and 10 g/(d·head), in the GOS2.5, GOS5, and GOS10 groups, respectively. GOS supplementation was provided for a duration of 28 days, starting from day 1 of birth.

### 2.2. Diet, Feeding, and Management

This study was carried out at Hebei Junyuan Dairy Farm (Xinle, China). GOS (≥90% purity) was purchased from Quantum hi-tech Biological Co., Ltd. (Guangzhou, China). The heifers were transferred to individual pens (1.8 × 1.4 × 1.2 m, enclosed by iron railings and lined with hay) immediately and received 4 L of colostrum within 1 h after birth. The heifers were fed 2 L of colostrum 3 times a day (06:00, 12:00, and 18:00) on days 2 and 3 after birth, followed by 8 L of heated raw milk from day 4 onwards. Appropriate doses of GOS were dissolved in a small quantity of colostrum/milk to ensure complete consumption before providing the remaining milk to the experimental groups during their daily morning feeding from 1 to 28 days of age. The starter was provided to the calves from 4 days of age. The remaining starter and the feed intake were recorded before morning feeding. Throughout this period, the heifers had ad libitum access to starter feed and water. The milk was produced by the farm, and it was sterilized by heating at 60 °C for 30 min. The starter was a Zhengda 970 (4–90 days old) concentrate supplement (Huhehaote, China). The feed was regularly checked to ensure it was within the established safety limits for animal consumption. The nutrient levels of the starter and milk are shown in Table 1.

### 2.3. Sample Collection and Analysis

#### 2.3.1. Starter and Milk

The starter was collected every week during the experimental period. The contents of dry matter (DM, AOAC International, 2005; method 930.15) [25], crude protein (CP, AOAC International, 2000; method 976.05) [26], and ether extract (EE, AOAC International, 2003; method 4.5.05) [27] in the starter were analyzed based on the standard methods described by the Association of Official Analytical Chemists (AOAC). The contents of acid detergent fiber and neutral detergent fiber were measured following Van Soest et al. [28].

The milk was collected in a 50 mL centrifuge tube before morning feeding every week and mixed with 1 pill of potassium dichromate preservative (Bronopol tablet, D&F Control System, San Ranmon Inc., Dublin, ON, Canada). It was then stored at −20 °C for milk composition analysis. The contents of milk protein, fat, lactose, total solids, and non-fat solids were measured using infrared analysis (Foss MilkoScan 2000, Foss Food Technology Corp., Eden Prairie, MN, USA).

#### 2.3.2. Growth Performance and Diarrhea Incidence

The initial body weight (IBW) and final body weight (FBW) of heifer calves were recorded on the day of birth and day 29 respectively. These values were used to calculate the average daily gain (ADG). The milk and starter intakes of heifers were recorded daily throughout the experimental period to determine the starter average daily feed intake (starter ADFI), total average daily feed intake (total ADFI), and feed efficiency (F/G).

The presence of diarrhea was assessed daily by the experimenter using the 4-point fecal scale [29] (Table 2). A fecal score ≥ 3 for two consecutive days or more was considered an indicator of diarrhea. Subsequently, the incidence of diarrhea was calculated according to the following formula:$$\mathrm{Diarrhea}\;\mathrm{incidence}\;\left(\%\right)=\frac{\mathrm{No}.\;\mathrm{of}\;\mathrm{calves}\;\mathrm{with}\;\mathrm{diarrhea}\;\mathrm{per}\;\mathrm{group}\times \;\mathrm{No}.\;\mathrm{of}\;\mathrm{days}\;\mathrm{of}\;\mathrm{diarrhea}}{\mathrm{No}.\;\mathrm{of}\;\mathrm{calves}\;\mathrm{per}\;\mathrm{group}\times \;\mathrm{No}.\;\mathrm{of}\;\mathrm{days}\;\mathrm{of}\;\mathrm{experiment}}\times 100\%$$

#### 2.3.3. Serum Sampling and Analysis

Blood samples were collected from heifers through the jugular vein and placed into a 10 mL non-anticoagulant vacuum tube on day 29 before the morning feeding. The collected blood was then centrifuged at 3000× *g* for 15 min at 4 °C. The serum was separated into 2 mL centrifuge tubes and stored at −20 °C for further analysis.

Immunoglobulins and inflammatory cytokines: Serum immunoglobulin (IgA, IgG, and IgM) and inflammatory cytokine (interleukin (IL-) 6, IL-1β, and IL-10) concentrations were tested using ELISA kits (Wuhan ColorfulGene Biotech Co., Ltd., Wuhan, China) according to manufacturer’s instructions.Biochemical indexes: Serum total protein (TP), total cholesterol (TC), triglyceride (TG), blood urea nitrogen (BUN), and glucose (GLU) concentrations were detected using an automatic biochemical analysis (Hitachi 7080, Tokyo, Japan).Appetite-related hormones: Serum leptin (LP), cholecystokinin (CCK), and glucagon-like peptide-1 (GLP-1) concentrations were tested using ELISA kits (Wuhan ColorfulGene Biotech Co., Ltd., Wuhan, China) according to the manufacturer’s instructions.Mineral elements: The serum contents of calcium (Ca), phosphorus (P), copper (Cu), iron (Fe), zinc (Zn), and magnesium (Mg) were determined using inductively coupled plasma optical emission spectroscopy (ICP-OES, PQ 9000, Analytik Jena, Jena, Germany), as described in the Chinese National Standards (GB 5009.268, China, 2016) and as reported previously [30].

### 2.4. Statistical Analysis

Data are expressed as the least square mean and standard error of the mean (SEM). Data analyses were performed using the Statistical Analysis System (SAS) 9.4 software, with preliminary collation performed using Microsoft Excel 2019. All variables have been verified for normality with the Shapiro–Wilk test and homoscedasticity with Levene’s test. The incidence of diarrhea was analyzed using the GENMOD model, and the effects of treatment on growth performance and serum indexes were analyzed using the GLM model. Multiple comparisons were conducted using Tukey’s method. Linear and quadratic multiple comparisons were used to evaluate the effects of GOS supplementation levels. *p* < 0.05 indicated statistically significant differences.

## 3. Results

### 3.1. Growth Performance and Incidence of Diarrhea

As shown in Table 3, the FBW, ADG, and feed efficiency of heifers linearly increased as the amounts of GOS supplementation increased (*p* < 0.05). Meanwhile, the starter ADFI and total ADFI quadratically increased and were higher in the GOS2.5 and GOS5 groups than in the CON group (*p* < 0.01).

### 3.2. Serum Immunoglobulins and Inflammatory Cytokines

As shown in Table 4, the serum concentrations of IgA linearly increased with increasing amounts of GOS (*p* < 0.05).

As shown in Table 5, the serum concentrations of IL-1β linearly decreased as the supplementation amount of GOS increased (*p* < 0.01). These values were significantly lower in all experimental groups than in the CON group (*p* < 0.01). The serum concentrations of IL-6 linearly declined with increasing GOS supplementation (*p* < 0.01), and they were significantly lower in the GOS5 and GOS10 groups than in the CON group (*p* < 0.01).

### 3.3. Serum Biochemical Indexes

As shown in Table 6, the serum concentrations of TG linearly declined as the supplementation of GOS increased (*p* < 0.01). However, serum TP and BUN levels linearly increased with increasing GOS supplementation (*p* < 0.01).

### 3.4. Serum Appetite-Related Hormones

As shown in Table 7, the serum concentrations of LP linearly reduced as the amount of GOS supplemented increased (*p* < 0.01). In contrast, the serum concentrations of GLP-1 and CCK linearly increased as the GOS dose increased (*p* < 0.01). 

### 3.5. Serum Mineral Elements

As shown in Table 8, the serum concentrations of Ca and Cu linearly increased as the amount of GOS supplementation increased (*p* < 0.01). The serum concentrations of Zn and Mg increased quadratically (*p* < 0.01), with both peaking in the GOS5 group (*p* < 0.05).

## 4. Discussion

Due to the prebiotic effects of GOS, it has been widely used as a natural supplement in young animals [31]. The results of our previous studies have shown that GOS has a good potential for use in calf rearing [21,24]. The purpose of this study is to explore even lower effective doses of GOS in calves. Furthermore, to better understand the effects of GOS on Holstein heifer calves, appetite-related hormones were also determined in the present study.

In the dairy farming industry, young cattle aged 0–6 months are referred to as calves. Young calves aged 0–2 months are considered pre-weaning calves, while those aged 3–6 months are considered post-weaning calves [32,33]. The ADG and ADFI are critical indicators of pre-weaning calf development Soberon et al. demonstrated that pre-weaning ADG significantly affects milk yield during the first lactation in the adult, with a 100 g higher ADG in calves corresponding to an 85–113 kg increase in milk production during lactation. Extensive research suggests that oligosaccharides can improve the growth performance of livestock, although most of these studies focus on monogastric animals [34,35,36,37]. For instance, one study showed that adding mannan oligosaccharides to the diet significantly increased the ADG in male broiler chickens aged 1–42 days [38]. Meanwhile, Craig et al. [39] found that oligosaccharide supplementation can significantly improve the feed conversion ratio in Ross-type broiler chickens. Additionally, GOS supplementation can significantly increase the ADFI of weaned piglets and reduce their diarrhea incidence [40]. The present study demonstrated that GOS supplementation can significantly increase the ADG and ADFI of Holstein calves and their feed efficiency (*p* < 0.05), which is consistent with the effects observed in monogastric animals [38,39,40]. There are β-galactosidase-encoding genes in the genomes of bifidobacterium species, which encode galactosidase synthesis with preferential hydrolysis of GOS [41]. GOS can be utilized and metabolized to SCFAs by intestinal microorganisms, serving as an energy source for colonic epithelial cells and thereby enhancing the nutrients available for host growth [42]. Furthermore, GO-mediated improvements in feed efficiency also appear to be related to the protective effect of GOS on intestinal mucosal barrier integrity.

Due to the unique structure of the bovine cotyledonary placenta, the transfer of immunoglobulins between the dam and the fetus is hindered, leading to low immunity in newborn calves [43,44]. Colostrum is rich in immunoglobulins, cytokines, oligosaccharides, and hormones. Thus, it is vital for improving disease resistance in newborn calves [45]. In the present study, heifer calves fed with different amounts of GOS showed significantly increased serum IgA concentrations, which may be related to intestinal health. IgA primarily serves to protect mucosal surfaces and reduce antigen invasion [46]. GOS is considered a bifidogenic factor, and it can selectively stimulate the proliferation of lactic acid bacteria and bifidobacteria in the digestive tract [47]. These bacteria can enhance local immune responses within the intestinal mucosa and stimulate B lymphocytes to release IgA, inhibiting and killing pathogens [46]. Consistent with our results, Lee et al. [48] found that GOS significantly increased the concentrations of IgA in sow colostrum during lactation and reduced the number of harmful bacteria in the faces of fed piglets. Moreover, Alizadeh et al. [49] also demonstrated that GOS can increase secretory IgA levels in the saliva of piglets. 

Pro-inflammatory cytokines, such as IL-1β and IL-6, are believed to impair the intestinal barrier by rearranging tight junction proteins, further inducing an inflammatory response [50]. Oligosaccharides down-regulate the expression of inflammatory factors in mesenteric lymphoid tissue to inhibit inflammation [51]. Cai et al. [52] showed that GOS can significantly reduce the IL-6 and IL-1β levels in the blood of calves at 4 and 6 weeks of age. Gao et al. [53] discovered that GOS can significantly reduce the IL-1β and IL-6 levels in LPS-challenged weaning piglets. In the present study, both serum IL-1β and IL-6 concentrations linearly decreased with an increase in the amount of supplemented GOS, which is consistent with these previous studies.

The present study showed that GOS supplementation not only improves the growth performance and immune functions in pre-weaning heifer calves but also has positive effects on their blood metabolites and mineral balance. In the present study, serum TG levels declined linearly with an increase in GOS supplementation. TG, a sterol, is a type of lipid that forms the cell membranes. It is synthesized by animal cells and serves as a precursor for bile acids, vitamin D, and steroid hormones. However, excessive TG levels in the body can trigger atherosclerosis and lipoprotein metabolism disorders [54]. Kong et al. [55,56] reported that GOS accelerates TG catabolism by promoting bile acid synthesis. Chen et al. [55] demonstrated the dose–dependent influence of GOS in regulating blood lipid levels in Sprague–Dawley rats. Although extensive research has shown that GOS effectively improves lipid profiles and lowers serum/plasma TC, TG, and LDL levels [57], the optimal dosage and timing of GOS administration for achieving cholesterol-lowering effects is unclear and warrants further research.

Serum TP content serves as a reference for assessing passive immunity in newborn calves. Our previous research has shown that supplementation with GOS significantly increases the serum concentrations of TP [21]. Similarly, in this study, both serum TP and BUN levels were found to linearly increase with increasing GOS supplementation. These effects might be linked to the regulatory effect of GOS on rumen function because GOS—as a fermentation substrate—increases the amount of organic matter supplied to rumen microorganisms, supporting the production of SCFAs and microbial proteins [58]. However, excessive GOS degradation may lead to a decrease in ruminal pH, affecting microbial activity and inhibiting microbial protein production in the rumen and subsequently influencing protein utilization in the body [59,60,61]. Therefore, selecting an appropriate dosage of GOS is the key to improving nitrogen utilization in calves.

Interestingly, GOS can also regulate lipid metabolism at the hormone level. LP is mainly secreted by white adipocytes, and its levels reflect the body’s fat content [62]. The stomach and intestines are the sources of LP and also contain LP receptors [63]. The circulating concentration of LP is negatively correlated with the host intestinal microbial diversity in the host [64]. Song et al. [65] demonstrated that in ob/ob mice, inulin can alleviate glucose and lipid metabolism disorders by improving the gut microbiota and microbial diversity and modulating LP metabolism pathways. It can also mediate dopamine signaling, which is associated with the hedonic response in feeding behavior [66]. This hormone-based perspective could explain the stimulatory effect of GOS on calf appetite. Rodenburg et al. [67] found that fructo-oligosaccharides (FOS) can induce the gene expression of *GLP-1*, *CCK*, and peptide YY in rats. These gut-derived hormones have been demonstrated to produce the intestinal epithelium in vivo [68]. Studies have also shown that supplementation with GOS increases the colonic expression of *GLP-1* in rats [69]. GLP-1 itself can inhibit the production of intestinal chylomicrons through the gut-brain axis [48]. CCK is a gastrointestinal hormone that induces gallbladder contraction [70]. Research shows that CCK affects intake behavior by mediating satiety pathways [71]. Accumulating evidence also suggests that SCFAs have broader systemic effects. Moreover, SCFAs produced by GOS fermentation can inhibit appetite by binding to free fatty acid receptors (FFAR2, GRP43), further activating the release of GLP-1, CCK, insulin, and LP and modulating the appetite system [72,73]. GOS shows a dose–dependent effect on feed intake and weight gain, as demonstrated in a clinical study where supplementation with at least 12 g of GOS per day significantly reduced food intake and body weight in overweight patients [74,75]. This may explain the significant increase in serum GLP-1 and CCK concentrations in the GOS10 group in the present study. In addition, investigating the effect of GOS on metabolism and energy level in calves could be beneficial for elucidating the exact mechanism in appetite regulation. Additionally, expanding the sample size would also be helpful.

Trace elements play an important role in stimulating and supporting the function of immunoglobulins. Ca enables the achievement of peak bone mass during growth in young animals and reduces bone loss in adulthood [76]. GOS can be fermented by colonic microbes, producing SCFAs and lowering intestinal pH, thereby increasing mineral solubility [77]. Furthermore, many intestinal microbial metabolites, including butyrate, can stimulate intestinal epithelial absorption. This could explain how GOS enhances mineral element absorption [78]. Our previous study demonstrated that GOS increases serum Ca concentrations in calves at 28 days of age, which is consistent with the findings of the present study. Cu serves as a cofactor for enzymes such as diamine oxidase, copper–zinc superoxide dismutase, and copper oxidase. Diamine oxidase is a highly active intracellular enzyme expressed in the villous layer of the small intestinal mucosa in mammals. Copper–zinc superoxide dismutase and copper oxidase are components of the body’s antioxidant system [79]. Tenorio et al. [15] found that GOS increases the absorption of Ca, Mg, and Cu and contributes to the apparent mineral balance. In the present study, serum Cu concentrations linearly increased as the amount of GOS supplemented increased, suggesting that GOS may modulate Cu absorption to alleviate physical and oxidative damage to the intestinal mucosal barrier in calves [80]. Notably, Mg is crucial for maintaining growth performance, and chronic Mg deficiency inhibits the growth and development of calves [81]. Bryk et al. [82] found that GOS and FOS consumption increase femur Ca, Mg, and P contents in growing rats. In the current study, the serum Mg concentration of heifers was the highest in the GOS5 group, suggesting that the optimal dose of GOS is 5 g/(d·head). Given that the rumen is the primary site for Mg absorption [83], these findings suggest a potential regulatory effect of GOS on Mg absorption in the rumen. This may suggest that further studies could focus on the effect of GOS on rumen development in pre-weaning dairy calves.

## 5. Conclusions

The present study demonstrated that supplementation with GOS significantly reduces the incidence of diarrhea and improves the growth performance of Holstein dairy calves. Meanwhile, it increases the serum concentrations of IgA and decreases inflammatory cytokines. Furthermore, GOS lowers serum TG levels and increases the serum Cu and Mg levels in dairy calves. In addition, supplementation with GOS affects serum appetite-related hormones by linearly reducing the concentration of LP and up-regulating GLP-1 and CCK in the serum of Holstein heifer calves. In summary, GOS supplementation is beneficial for reducing the incidence of diarrhea and improving the growth performance and immune function of Holstein heifer calves. These effects can be attributed to the facilitatory effect of GOS on nutrient absorption and appetite hormone-dominated feeding activities. Under the conditions of the present study, supplementation with 5 g/(d·head) GOS to newborn Holstein heifers is recommended. The findings of this study provide new insights into the rational application of GOS in calf rearing. Further studies are still needed to investigate the optimal associative effects of the feeding dose and duration of GOS on dairy calves during their early life due to their persistent functions.

## Figures and Tables

**Table 1 animals-13-03366-t001:** Nutrient levels of the starter (based on dry matter) and milk.

Items	Starter	Items	Milk
Dry Matter (%)	88.93	Milk protein (%)	3.74
Crude Protein (% DM)	24.19	Milk fat (%)	4.35
Ether Extract (% DM)	1.60	Lactose (%)	5.00
Acid Detergent Fiber (% DM)	11.24	Total solids (%)	13.79
Neutral Detergent Fiber (% DM)	23.55	Non-fat solids (%)	9.44
Ash (% DM)	9.05	Density (g/mL)	1.04

**Table 2 animals-13-03366-t002:** Fecal 4-point scale.

Appearance	Mobility	Scores
Normal	Firm but not hard	1
Soft	Piles but spreads slightly	2
Runny	Spreads readily	3
Watery	Liquid consistency, splatters	4

**Table 3 animals-13-03366-t003:** Effects of different levels of galacto-oligosaccharides (GOS) on the growth performance and incidence of diarrhea in Holstein heifer calves during 1–28 days of treatment.

Items	Treatment ^1^				SEM ^2^	*p*-Value		
	CON	GOS2.5	GOS5	GOS10		Treatment	Linear	Quadratic
IBW ^3^ (d 1, kg)	41.29	40.93	41.21	41.29	0.36	0.98	0.91	0.83
FBW ^4^ (d 29, kg)	58.14 ^b^	58.86 ^ab^	60.86 ^ab^	61.43 ^a^	0.47	0.03	<0.01	0.43
ADG ^5^ (g/d)	602.04 ^c^	640.31 ^bc^	701.53 ^ab^	719.39 ^a^	12.01	<0.01	<0.01	0.09
Starter ADFI ^6^ (g DM/d)	102.46 ^b^	139.05 ^a^	132.73 ^a^	100.15 ^b^	3.80	<0.01	0.20	<0.01
Total ADFI (g DM/d)	1249.79 ^b^	1286.38 ^a^	1280.06 ^a^	1247.48 ^b^	3.80	<0.01	0.20	<0.01
Feed efficiency (g of DMI/g of gain)	2.00 ^a^	1.93 ^ab^	1.75 ^bc^	1.67 ^c^	0.04	<0.01	<0.01	0.06
Incidence of diarrhea (%)	13.27	10.71	10.2	11.73	-	0.79	-	-

^1^ CON, the control group; GOS2.5, GOS5, GOS10, the experimental groups in which calves received 2.5, 5, 10 g/(d·head) GOS, respectively; ^2^ SEM, standard error of the mean; ^3^ IBW, initial body weight; ^4^ FBW, final body weight; ^5^ ADG, average daily gain; ^6^ ADFI, average daily feed intake. ^a,b,c^ Values denote significant differences within the row (*p* < 0.05).

**Table 4 animals-13-03366-t004:** Effects of different levels of galacto-oligosaccharides (GOS) on serum immunoglobulin concentrations of Holstein heifer calves.

Items	Treatment ^1^				SEM ^2^	*p*-Value		
	CON	GOS2.5	GOS5	GOS10		Treatment	Linear	Quadratic
IgA ^3^ (μg/mL)	45.91	46.88	47.78	50.34	0.72	0.15	0.02	0.85
IgG ^4^ (mg/mL)	20.35	20.69	21.55	21.43	0.30	0.45	0.19	0.46
IgM ^5^ (mg/mL)	17.97	17.62	18.11	18.10	0.13	0.57	0.49	0.80

^1^ CON, the control group; GOS2.5, GOS5, GOS10, the experimental groups in which calves received 2.5, 5, 10 g/(d·head) GOS, respectively; ^2^ SEM, standard error of the mean; ^3^ IgA, immunoglobulin A; ^4^ IgG, immunoglobulin G; ^5^ IgM, immunoglobulin M.

**Table 5 animals-13-03366-t005:** Effects of different levels of galacto-oligosaccharides (GOS) on serum cytokine concentrations in Holstein heifer calves (pg/mL).

Items	Treatment ^1^				SEM ^2^	*p*-Value		
	CON	GOS2.5	GOS5	GOS10		Treatment	Linear	Quadratic
Interleukin-6	143.91 ^a^	134.79 ^ab^	127.58 ^b^	125.41 ^b^	2.30	<0.01	<0.01	0.10
Interleukin-1β	214.24 ^a^	194.38 ^b^	189.71 ^b^	183.12 ^b^	3.44	<0.01	<0.01	0.06
Interleukin-10	244.44	247.79	269.56	258.56	3.64	0.05	0.07	0.08

^1^ CON, the control group; GOS2.5, GOS5, GOS10, the experimental groups in which calves received 2.5, 5, 10 g/(d·head) GOS, respectively; ^2^ SEM, standard error of the mean. ^a,b^ Values denote significant differences within the row (*p* < 0.05).

**Table 6 animals-13-03366-t006:** Effects of different levels of galacto-oligosaccharides (GOS) on the serum indexes of Holstein heifer calves.

Items	Treatment ^1^				SEM ^2^	*p*-Value		
	CON	GOS2.5	GOS5	GOS10		Treatment	Linear	Quadratic
TC (mmol/L)	2.48	2.57	2.86	2.88	0.14	0.72	0.31	0.73
TG (mmol/L)	0.43 ^a^	0.34 ^ab^	0.27 ^b^	0.24 ^b^	0.02	0.01	<0.01	0.16
TP (g/L)	51.95 ^b^	54.24 ^ab^	57.90 ^ab^	59.93 ^a^	1.14	0.03	<0.01	0.49
BUN (mmol/L)	3.04 ^b^	3.49 ^ab^	3.83 ^ab^	3.91 ^a^	0.12	0.03	<0.01	0.15
GLU (mmol/L)	6.77	6.63	6.26	6.27	0.12	0.33	0.16	0.82

^1^ CON, the control group; GOS2.5, GOS5, GOS10, the experimental groups in which calves received 2.5, 5, 10 g/(d·head) GOS, respectively; ^2^ SEM, standard error of the mean; TC, total cholesterol; TG, triglyceride; TP, total protein; BUN, blood urea nitrogen; GLU, glucose. ^a,b^ Values denote significant differences within the row (*p* < 0.05).

**Table 7 animals-13-03366-t007:** Effects of different levels of galacto-oligosaccharides (GOS) on serum appetite-related hormone concentrations in Holstein heifer calves (pg/mL).

Items	Treatment ^1^				SEM ^2^	*p*-Value		
	CON	GOS2.5	GOS5	GOS10		Treatment	Linear	Quadratic
Leptin	2367.24 ^a^	2354.41 ^ab^	2342.65 ^ab^	2292.06 ^b^	10.24	0.03	<0.01	0.56
Cholecystokinin	196.76	200.73	201.22	205.85	1.28	0.08	0.01	0.83
Glucagon-like peptide-1	274.38 ^b^	291.62 ^ab^	294.05 ^ab^	312.55 ^a^	4.55	0.02	<0.01	0.70

^1^ CON, the control group; GOS2.5, GOS5, GOS10, the experimental groups in which calves received 2.5, 5, 10 g/(d·head) GOS, respectively; ^2^ SEM, standard error of the mean. ^a,b^ Values denote significant differences within the row (*p* < 0.05).

**Table 8 animals-13-03366-t008:** Effects of different levels of galacto-oligosaccharides (GOS) on serum trace element concentrations in Holstein heifer calves (mg/L).

Items	Treatment ^1^				SEM ^2^	*p*-Value		
	CON	GOS2.5	GOS5	GOS10		Treatment	Linear	Quadratic
Ca	99.02 ^b^	102.82 ^b^	103.88 ^ab^	108.06 ^a^	1.21	0.05	<0.01	0.74
P	116.56	117.68	126.16	118.74	1.83	0.24	0.54	0.13
Cu	0.75 ^c^	0.85 ^bc^	1.02 ^ab^	1.07 ^a^	0.04	<0.01	<0.01	0.15
Fe	1.25	1.54	1.71	1.53	0.07	0.17	0.21	0.06
Zn	1.08	1.12	1.23	1.14	0.02	0.09	0.27	0.04
Mg	18.48 ^b^	18.54 ^ab^	19.44 ^a^	18.58 ^ab^	0.14	0.03	0.55	0.02

^1^ CON, the control group; GOS2.5, GOS5, GOS10, the experimental groups in which calves received 2.5, 5, 10 g/(d·head) GOS, respectively; ^2^ SEM, standard error of the mean. Ca, calcium; P, phosphorus; Cu, copper; Fe, iron; Zn, zinc; and Mg, magnesium. ^a,b,c^ Values denote significant differences within the row (*p* < 0.05).

## Data Availability

The data presented in this study are available in the article.

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
