# Peer review of "Effects of Different Galacto-Oligosaccharide Supplementation on Growth Performance, Immune Function, Serum Nutrients, and Appetite-Related Hormones in Holstein Calves"

_animals, 2023, doi:10.3390/ani13213366_

Round 1

Reviewer 1 Report

Comments and Suggestions for Authors

The research, titled “Effects of Different Levels of Galacto-Oligosaccharides on Growth Performance, Immune Function, Serum Nutrients and Appetite-Related Hormones of Holstein Heifer Calves“ addresses an important and timely topic. I found the subject matter of the article fascinating and read the manuscript with great interest. The paper aligns well with the scope of the journal. However, I believe that in its current form, it has several shortcomings.

This study aimed to investigate the effects of galacto-oligosaccharides (GOS) supplementation on the growth, immune function, serum nutrient levels, and appetite-related hormones in newborn Holstein heifer calves. The results demonstrated that GOS supplementation, particularly at 5 g/(d·head), significantly improved growth performance, enhanced immune function, and positively influenced serum nutrient levels in these calves. Notably, GOS showed promise in regulating nutrient absorption and microelement levels. This research highlights the potential of GOS as a beneficial dietary supplement for Holstein heifer calves, with a recommended dose of 5 g/(d·head).

Specific comments:

I suggest rewriting the simple summary. According to the author's guidelines, this section should summarize and contextualize your paper within the existing literature in your field. It should be written without technical language or nonstandard acronyms, with the goal of conveying the meaning and importance of this research to non-experts

I recommend rewriting the abstract and including more results and the significance of the obtained data.

Introduction:

The introduction lacks an extensive literature review, especially concerning GOS supplementation in calf nutrition. A more comprehensive overview of existing research on GOS and its effects on calf health and growth would provide better context for the study.

Line 58: please consider to cite: 10.1016/j.rvsc.2023.03.008

While the study aims to investigate the effects of GOS supplementation, the specific hypothesis or research questions should be explicitly stated in the introduction. This will help readers understand the study's objectives more clearly.

Methods:

The methods section could benefit from greater detail. For instance, it would be helpful to include information on the source of GOS, the method of administration, and any potential variations in calf management that might have affected the results.

I suggest expanding the Methods section to provide a more detailed and comprehensive description of the procedures. This will enhance the clarity and replicability of your study. Consider including the following details:

The sample size of 28 heifer calves is relatively small. A more extensive sample would strengthen the study's statistical power and make the results more representative. Additionally, it's essential to report the representativeness of the chosen calves in terms of breed, age, and health status.

Did you assess the body condition score (BCS) of the cattle in your study? If so, please report the BCS results and provide an appropriate reference for the BCS assessment method you used. Consider citing: 10.1080/1828051X.2022.2032850.

To enhance the transparency and replicability of your research, I kindly suggest that you include a section detailing the methods employed for dietary analysis. This should encompass the techniques and procedures used to determine the composition of the diet. Furthermore, I recommend referencing a reputable source for these methods, such as the protocol outlined in 10.3390/ani13050797 and 10.3390/ani12141740.

The potential impact of aflatoxin levels in the feed, particularly in the corn used during the trial, on liver function and study outcomes is a valid concern. To address this issue and ensure the integrity of our study, we would like to confirm that rigorous quality control measures were implemented throughout the study. Specifically, the feed provided to the animals, including the corn, was regularly tested to ensure that aflatoxin levels remained well below established safety limits for animal consumption. This stringent monitoring was undertaken to mitigate any potential bias related to aflatoxin contamination, which could adversely affect liver health and consequently influence the study's results.

Please report a specific comment regarding the absence of this kind of bias of your study such as: "The diet provided in this study was carefully monitored to ensure that aflatoxin levels were well below the established safety limits for animal feed. This precautionary measure was taken to safeguard the animals' health and welfare. Aflatoxin contamination in animal feed can pose serious health risks, including impaired growth and liver damage (see, for example, 10.3390/toxins14070430). By maintaining feed quality within safe limits, we aimed to minimize any potential influence of aflatoxins on the study results."

Could you please clarify whether you conducted tests for normality and homogeneity on your data before proceeding with the statistical analysis? It's crucial to ensure that the assumptions underlying your chosen statistical methods are met. I recommend referring to the guidelines outlined in [proposed reference, e.g., 10.1080/1828051X.2020.1827990] for conducting such tests to maintain the rigor and reliability of your analysis.

Explain how the data were presented and whether any transformations or adjustments were made to the raw data. Clarify how outliers, if any, were handled in the analysis.

To facilitate transparency and future research, consider sharing the data and detailed methodology used in this study.

Discussion:

Starting the discussion section by reiterating the aim of the study can provide clarity and context for readers.

The discussion section could expand on the practical implications of the findings. How might GOS supplementation be practically implemented in calf rearing, and what benefits could it offer to farmers and the dairy industry?

It would be valuable to include an assessment of feed palatability in your study. Feed palatability is a critical factor influencing feed intake and, subsequently, animal performance. It can significantly affect the acceptance and consumption of specific feed components. Adding a section discussing feed palatability and citing relevant references in animal feeding practice would enhance the comprehensiveness of your study. Consider to cite: 10.1016/j.applanim.2020.105110

To strengthen the study and provide a more comprehensive context for the growth performance results, it would indeed be valuable to compare them with a wider range of relevant literature. By including a more extensive review of existing research on similar topics, the study can better demonstrate how its findings fit into the broader body of knowledge in calf nutrition and growth. This would not only enhance the power of the study but also provide readers with a more comprehensive understanding of the significance of the results. Consider to cite: 10.3390/vetsci10090554

Considering the potential economic implications of GOS supplementation is crucial, especially for farmers. An economic analysis section discussing the cost-effectiveness and potential returns on investment for GOS supplementation would enhance the study's practical applicability.

Explicitly addressing the limitations of the study, such as the small sample size and potential confounding factors, would provide a more balanced view of the research. Suggesting directions for future research, such as larger-scale trials or long-term effects, would be valuable.

Conclusion:

I kindly suggest expanding the conclusions section of your paper to provide a more detailed and comprehensive report of the main findings. This will help readers better understand the significance of your research.

Please double-check the reference list to ensure that all references are included in the main text and vice versa.

Overall, this study provides valuable insights into GOS supplementation for calf nutrition, but addressing these points would enhance its clarity, relevance, and practical applicability in the field of animal husbandry.

Author Response

Reviewer: 1

Comments and Suggestions for Authors:

The research, titled “Effects of Different Levels of Galacto-Oligosaccharides on Growth Performance, Immune Function, Serum Nutrients and Appetite-Related Hormones of Holstein Heifer Calves’’ addresses an important and timely topic. I found the subject matter of the article fascinating and read the manuscript with great interest. The paper aligns well with the scope of the journal. However, I believe that in its current form, it has several shortcomings.

This study aimed to investigate the effects of galacto-oligosaccharides (GOS) supplementation on growth, immune function, serum nutrient levels, and appetite-related hormones in newborn Holstein heifer calves. The results demonstrated that GOS supplementation, particularly at 5 g/(d·head), significantly improved growth performance, enhanced immune function, and positively influenced serum nutrient levels in these calves. Notably, GOS showed promise in regulating nutrient absorption and microelement levels. This research highlights the potential of GOS as a beneficial dietary supplement for Holstein heifer calves, with a recommended dose of 5 g/(d·head).

Response: Thank you for your valuable comments. The manuscript has been revised according to your comments. All the changes have been highlighted in yellow colour in the revised manuscript. Thank you for your time and patience.

Specific comments:

  1. I suggest rewriting the simple summary. According to the author's guidelines, this section should summarize and contextualize your paper within the existing literature in your field. It should be written without technical language or nonstandard acronyms, with the goal of conveying the meaning and importance of this research to non-experts.

I recommend rewriting the abstract and including more results and the significance of the obtained data.

Response: Thank you for your valuable suggestions. We have rewritten these two parts of this article as you suggested. Please check the highlighted section in the attached manuscript. (Line 12–18, Line 19–50)

-

Introduction:

  1. The introduction lacks an extensive literature review, especially concerning GOS supplementation in calf nutrition. A more comprehensive overview of existing research on GOS and its effects on calf health and growth would provide better context for the study.

Response: Thank you for your valuable comments. We have added some research results regarding GOS in calves. Meanwhile, we have rewritten the introduction as you suggested, please check the highlighted section in the attached manuscript. (Line 85–97)

  1. Line 58: please consider to cite: 10.1016/j.rvsc.2023.03.008

Response: Thank you for your reminder, we have cited the appropriate references here. (Line 61)

  1. While the study aims to investigate the effects of GOS supplementation, the specific hypothesis or research questions should be explicitly stated in the introduction. This will help readers understand the study's objectives more clearly.

Response: Thank you for your valuable suggestion, we have stated the hypothesis and purpose of this study. (Line 100–108)

Methods:

  1. The methods section could benefit from greater detail. For instance, it would be helpful to include information on the source of GOS, the method of administration, and any potential variations in calf management that might have affected the results.

Response: Thank you for your valuable comments. The GOS used in this experiment was a commercial product, which was produced by Quantum hi-tech Biological Co. Ltd. (Guangzhou, China). The details of feeding procedures and calf management have been described in sections ‘2.1 Experimental Animals and Design’ and ‘2.2 Diet, Feeding, and Management’. Please check the attached manuscript. (Line 118–142)

I suggest expanding the Methods section to provide a more detailed and comprehensive description of the procedures. This will enhance the clarity and replicability of your study. Consider including the following details:

  1. The sample size of 28 heifer calves is relatively small. A more extensive sample would strengthen the study's statistical power and make the results more representative. Additionally, it's essential to report the representativeness of the chosen calves in terms of breed, age, and health status.

Response: Thank you for your valuable comments. We agree with you that the sample size of 28 heifer calves is relatively small. However, given the dose-response design of treatments, the use of linear and quadratic contrasts, along with treatments vs. control contrast, can be used to provide a more robust and powerful analysis of the results. Furthermore, we selected the sample size based on the effect size observed in our previous research. Thank you for your valuable suggestions, we would consider increasing the sample size in the design of subsequent experiments.

Thank you for your valuable comments. The breed of experimental calves are all newborn Holstein heifer calves. These calves with similar dates of birth and birth weights (41.18±1.90 kg) (mean±SD). The period of the experiment was from birth to 28 days of age. (Line 118–125)

  1. Did you assess the body condition score (BCS) of the cattle in your study? If so, please report the BCS results and provide an appropriate reference for the BCS assessment method you used. Consider citing: 10.1080/1828051X.2022.2032850.

Response: Thank you for your question and suggestion. In our study, we did not assess the Body Condition Score (BCS) of the cattle. Our research focused on investigating the growth, immune function, and mineral absorption of the calves, as described in the manuscript. But we think highly of your comments. We have noted the reference for the BCS assessment method you provided and will consider incorporating this method into our future research to gain a more comprehensive understanding of the health status of the cattle.

  1. To enhance the transparency and replicability of your research, I kindly suggest that you include a section detailing the methods employed for dietary analysis. This should encompass the techniques and procedures used to determine the composition of the diet. Furthermore, I recommend referencing a reputable source for these methods, such as the protocol outlined in 3390/ani13050797 and 10.3390/ani12141740.

Response: Thank you for your valuable comments. The contents of the starter were analyzed based on the standard methods described by the Association of Official Analytical Chemists (AOAC), and we have listed the references at the end of this article (Line 146–150). The details of the dietary analysis are described in sections ‘2.3.1 Starter and milk’. Please check the attached manuscript. (Line 145)

  1. The potential impact of aflatoxin levels in the feed, particularly in the corn used during the trial, on liver function and study outcomes is a valid concern. To address this issue and ensure the integrity of our study, we would like to confirm that rigorous quality control measures were implemented throughout the study. Specifically, the feed provided to the animals, including the corn, was regularly tested to ensure that aflatoxin levels remained well below established safety limits for animal consumption. This stringent monitoring was undertaken to mitigate any potential bias related to aflatoxin contamination, which could adversely affect liver health and consequently influence the study's results.

Please report a specific comment regarding the absence of this kind of bias of your study such as: "The diet provided in this study was carefully monitored to ensure that aflatoxin levels were well below the established safety limits for animal feed. This precautionary measure was taken to safeguard the animals' health and welfare. Aflatoxin contamination in animal feed can pose serious health risks, including impaired growth and liver damage (see, for example, 10.3390/toxins14070430). By maintaining feed quality within safe limits, we aimed to minimize any potential influence of aflatoxins on the study results."

Response: Thank you for your valuable comments. We appreciate your diligence in ensuring the integrity of our research. We have stated in the part of the 2.2 Diet, Feeding, and Management (Line 126). During the course of our study, we took stringent measures to maintain the quality of the feed, including the corn, provided to the animals. The feed was regularly checked to ensure that it remained within the established safety limits for animal consumption. Moreover, we strictly adhered to the expiration dates of the feed and ensured timely replacements when necessary. We acknowledge your concern and would like to assure you that we prioritized the welfare of the animals and the validity of our results throughout the research process.

  1. Could you please clarify whether you conducted tests for normality and homogeneity on your data before proceeding with the statistical analysis? It's crucial to ensure that the assumptions underlying your chosen statistical methods are met. I recommend referring to the guidelines outlined in [proposed reference, e.g., 1080/1828051X.2020.1827990] for conducting such tests to maintain the rigor and reliability of your analysis.

Response: Thank you for your valuable comments. We have conducted tests for normality and homogeneity to ensure that the assumptions underlying our chosen statistical methods are met. We have added this explanation in the section of ‘2.4 Statistical Analysis’ in yellow. (Line 195–197)

  1. Explain how the data were presented and whether any transformations or adjustments were made to the raw data. Clarify how outliers, if any, were handled in the analysis.

To facilitate transparency and future research, consider sharing the data and detailed methodology used in this study.

Response: Thank you for your valuable comments. All variables have been verified for normality with the Shapiro–Wilk’s test, and for homoscedasticity with the Levene's test. We would be glad to explain the abnormal data outside the range of (mean ± 2*SD) have been excluded. Furthermore, we encourage interested researchers to reach out to us for further information on the methods and data processing for potential collaboration or verification. We hope this explanation clarifies our decisions and assures you that our study strives to balance transparency while maintaining the integrity and privacy of the data. Once again, we appreciate your review and valuable feedback.

Discussion:

  1. Starting the discussion section by reiterating the aim of the study can provide clarity and context for readers.

Response: Thank you for your valuable suggestions, we have reiterated the purpose at the beginning of the discussion as you suggested. Please check the highlighted section in the attached manuscript. (Line 256–261)

  1. The discussion section could expand on the practical implications of the findings. How might GOS supplementation be practically implemented in calf rearing, and what benefits could it offer to farmers and the dairy industry?

Response: Thank you for your valuable suggestions, we stated the practical implications and significance of this study as you suggested, and we added it at the end of the conclusion. (Line 395–399)

  1. It would be valuable to include an assessment of feed palatability in your study. Feed palatability is a critical factor influencing feed intake and, subsequently, animal performance. It can significantly affect the acceptance and consumption of specific feed components. Adding a section discussing feed palatability and citing relevant references in animal feeding practice would enhance the comprehensiveness of your study. Consider to cite: 10.1016/j.applanim.2020.105110

Response: We appreciate your diligence in ensuring the integrity of our research. We will consider adding a section on feed value assessment of palatability to future studies. Thank you for your valuable comments and interesting insights.

  1. To strengthen the study and provide a more comprehensive context for the growth performance results, it would indeed be valuable to compare them with a wider range of relevant literature. By including a more extensive review of existing research on similar topics, the study can better demonstrate how its findings fit into the broader body of knowledge in calf nutrition and growth. This would not only enhance the power of the study but also provide readers with a more comprehensive understanding of the significance of the results. Consider to cite: 10.3390/vetsci10090554

Response: We sincerely appreciate your valuable feedback and the suggestion to incorporate a broader range of literature into the context of our study. In our research, we aimed to conduct a focused investigation into the effects of GOS supplementation on newborn Holstein heifer calves. We believe that our current contextualization already provides a comprehensive understanding of the significance of our results. We have thoroughly reviewed the relevant literature that pertains to our specific research question, which forms the basis for the rationale and hypothesis of our study.

  1. Considering the potential economic implications of GOS supplementation is crucial, especially for farmers. An economic analysis section discussing the cost-effectiveness and potential returns on investment for GOS supplementation would enhance the study's practical applicability.

Response: We appreciate your insightful suggestion regarding the potential economic implications of GOS supplementation in calf rearing. We have also paid attention to this problem when we originally designed our series of experiments concerning GOS. To our knowledge, the economic analysis includes not only the supplement dose, but also the supplementation duration. Therefore, in the present study, we aimed to find the appropriate dose effects of GOS, and we will investigate the appropriate supplementation duration, for example, for 7, 14 or 21 days, if possible, in the next experiment. After we have got both the optimal dose with the duration, that is, how much and how long of the usage of GOS for young calves, we will do the whole economic analysis for dairy production. We genuinely appreciate your suggestion.

  1. Explicitly addressing the limitations of the study, such as the small sample size and potential confounding factors, would provide a more balanced view of the research. Suggesting directions for future research, such as larger-scale trials or long-term effects, would be valuable.

Response: Thank you for your valuable comments. After comprehensively evaluating this article, we added a discussion on the limitations of this study and the baseline for future research. (Line 355–357, Line 381–383)

Conclusion:

  1. I kindly suggest expanding the conclusions section of your paper to provide a more detailed and comprehensive report of the main findings. This will help readers better understand the significance of your research.

Response: Thank you for your valuable comments. We have rewritten the conclusion as suggested. (Line 395–399)

  1. Please double-check the reference list to ensure that all references are included in the main text and vice versa.

Response: Thank you for your valuable comments. We have checked the references through the full article. 

Overall, this study provides valuable insights into GOS supplementation for calf nutrition, but addressing these points would enhance its clarity, relevance, and practical applicability in the field of animal husbandry.

Response: We think highly of your valuable comments and suggestions. We have revised our manuscript as you suggested. Thank you for your time and patience.

Reviewer 2 Report

Comments and Suggestions for Authors

Manuscript ID: animals-2625743

Type of Manuscript: Article

Title: Effects of Different Levels of Galacto-Oligosaccharides on Growth Performance, Immune Function, Serum Nutrients, and Appetite-Related Hormones of Holstein Heifer Calves

Authors: Xin Yu, Fengtao Ma, Haonan Dai, Junhao Liu, Nesrein M Hashem, Peng Sun*

The authors aim to investigate whether similar effects of GOS on growth performance, immune function, serum nutrients, and appetite-related hormones still exist in newborn Holstein heifer calves. While the present work provides interesting data, several concerns were identified throughout the manuscript that require the authors to make improvements before further consideration. Specific comments are provided below:

Major Concerns:

Since the present work is a continuation of previous research, it is essential to clarify the novelty of the current study compared to the earlier work. The significance of the research problem should be addressed, and it should be explained how the present work contributes to solving this problem.

Simple Summary:

There is an excessive level of detail regarding methodology and treatment in this section. Authors are encouraged to provide a concise summary of the research, and excessive treatment details should be avoided.

It should be clearly defined how the present work bridges the gap with the previous research.

Some abbreviations, such as AGD, should be defined upon first use.

Abstract:

Lines 20-21: Explain how the present work addresses the gap left by previous investigations. Highlight the new knowledge contributed.

Lines 30-31: Specify what the comparison is made against. Is it against a control group? Please clarify.

Lines 40-41: Specify the basis of comparison. Is it against a control group? Please clarify.

Introduction:

Lines 80-81: Provide more details on the animal feeding process.

Lines 82-84: Explain the prevalence of mineral deficiencies in young animals and how GOS supplementation can enhance mineral absorption in greater detail. Provide evidence for these claims.

Line 84 and others: Consider removing citations for items 11-16 as they are mentioned briefly in a single sentence, which doesn't provide much context.

Lines 90-95: Defend the novelty of the present work in comparison to your previous research. Clearly state the research gap.

Specify the levels of GOS used and how these levels were determined.

Materials and Methods:

Line 108: Provide evidence for the age of the animals.

Line 111: Define "Normal milk."

In section 2.3.3 ("Serum Sampling and Analysis"), consider rephrasing to improve clarity.

Results:

Line 185: Correct "Starter ADFI" to "starter ADFI."

In Table 3, explain why the incidence of diarrhea (%) does not have SEM reported.

Discussion:

Line 251: Explain how GOS can be utilized and metabolized.

Lines 268-270: Clarify if the findings can be compared to ruminant nutrition.

Lines 328-330: Ensure that these sentences are closely related to the context of the discussion.

Conclusions:

Recommend further research in this area.

References:

Verify the formatting of your references.

Comments on the Quality of English Language

Moderate editing of the English language is required.

Author Response

Reviewer: 2

Comments and Suggestions for Authors:

The authors aim to investigate whether similar effects of GOS on growth performance, immune function, serum nutrients, and appetite-related hormones still exist in newborn Holstein heifer calves. While the present work provides interesting data, several concerns were identified throughout the manuscript that require the authors to make improvements before further consideration. Specific comments are provided below:

Major Concerns:

  1. Since the present work is a continuation of previous research, it is essential to clarify the novelty of the current study compared to the earlier work. The significance of the research problem should be addressed, and it should be explained how the present work contributes to solving this problem.

Response: Thank you for your valuable comments. Our previous study had verified that early supplementation with 10 g/(d·head) of GOS in newborn Holstein dairy calves reduced the incidence of diarrhea and improved growth performance and mineral absorption. To furtherly reduce the feed cost, we want to furtherly investigate whether even lower dose of GOS can exhibit similar effects. The novelty of the current study compared to the earlier work has been explained in the abstract (Line 21–24). Thank you for your time and patience.

Simple Summary:

  1. There is an excessive level of detail regarding methodology and treatment in this section. Authors are encouraged to provide a concise summary of the research, and excessive treatment details should be avoided.

It should be clearly defined how the present work bridges the gap with the previous research.

Response: Thank you for your valuable suggestions, we have rewritten this simple summary of this article as you suggested. And explained the relationship between the present and the previous research in the abstract. Please check the highlighted section in the attached manuscript (Line 12–18, Line 19–50).

  1. Some abbreviations, such as AGD, should be defined upon first use.

Response: Thank you for your valuable comments. I suspect you may be referring to ADG, and we have indicated the full name of the abbreviation in yellow. Meanwhile, we have checked the abbreviations through the full text, as suggested. (Line 162)

Abstract:

  1. Lines 20-21: Explain how the present work addresses the gap left by previous investigations. Highlight the new knowledge contributed.

Response: Thank you for your valuable suggestions, we have rewritten the abstract part of this article as you suggested. (Line 21–24)

  1. Lines 30-31: Specify what the comparison is made against. Is it against a control group? Please clarify.

Response: Thank you for your valuable comments. This means the effect of treatment is in a significant linear increasing trend as the dose increases. (Line 30–31)

  1. Lines 40-41: Specify the basis of comparison. Is it against a control group? Please clarify.

Response: Thank you for your valuable comments. This means the effect of treatment is in a significant linear increasing trend as the dose increases. (Line 41–42)

Introduction:

  1. Lines 80-81: Provide more details on the animal feeding process.

Response: Thank you for your valuable comments. We have supplemented more details as you suggested. Please check the attached manuscript. (Line 85–89)

  1. Lines 82-84: Explain the prevalence of mineral deficiencies in young animals and how GOS supplementation can enhance mineral absorption in greater detail. Provide evidence for these claims.

Response: Thank you for your valuable comments. Considering the logic of the article, we mainly present the current research status of GOS in the introduction and explain the potential mechanisms in the discussion section. And we have written this in the part of the discussion. (Line 361–365)

  1. Line 84 and others: Consider removing citations for items 11-16 as they are mentioned briefly in a single sentence, which doesn't provide much context.

Response: We sincerely appreciate your valuable feedback and suggestions. We have thoroughly reviewed these literatures, which forms the basis for the rationale and hypothesis of our study. And we think it’s valuable to provide a comprehensive context about the function of GOS.

  1. Lines 90-95: Defend the novelty of the present work in comparison to your previous research. Clearly state the research gap.

Response: Thank you for your valuable comments. We have rewritten the introduction part of this article as you suggested. Please check the highlighted section in the attached manuscript. (Line 100–108)

  1. Specify the levels of GOS used and how these levels were determined.

Response: Thank you for your valuable comments. We specify these in the introduction section as you suggested. (Line 102)

Materials and Methods:

  1. Line 108: Provide evidence for the age of the animals.

Response: Thank you for your comments. The period of the experiment was from birth to 28 days of age.

  1. Line 111: Define "Normal milk."

Response: Thank you for your comments. To prevent barriers to understanding, we have changed it into ‘heated raw milk’. ‘heated raw milk’ is raw milk first heated to 42℃, and then naturally cooled to 38℃ before fed to calves.

  1. In section 2.3.3 ("Serum Sampling and Analysis"), consider rephrasing to improve clarity.

Response: Thank you for your valuable comments. We've polished the language to make it easy to read, as suggested. (Line 172)

Results:

  1. Line 185: Correct "Starter ADFI" to "starter ADFI."

Response: Thank you for your reminder. We have revised the abbreviations through the full text.

  1. In Table 3, explain why the incidence of diarrhea (%) does not have SEM reported.

Response: Thank you for your valuable comments. Since the incidence of diarrhea is a categorical variable, we used the chi-square test to do statistical analysis. The incidence of diarrhea (%) does not have SEM reported, because it is not a continuous variable.

Discussion:

  1. Line 251: Explain how GOS can be utilized and metabolized.

Response: Thank you for your valuable comments. We have explained the mechanisms of it as you suggested. (Line 277–281)

  1. Lines 268-270: Clarify if the findings can be compared to ruminant nutrition.

Response: Thank you for your valuable comments. Due to the rumen of calves (0–8 weeks of age) not fully developed, they rely on the crumpled stomach to play the main digestive function. Their digestive characteristics are similar to those of monogastric animals. Therefore, we think it is meaningful to list this reference about piglets here. Additionally, we have supplemented more research on calves in the introduction.

  1. Lines 328-330: Ensure that these sentences are closely related to the context of the discussion.

Response: Thank you for your valuable comments. Since GOS supplementation can enhance mineral absorption, we believe it could be helpful in explaining the context of this research’s motivation.

Conclusions:

  1. Recommend further research in this area.

Response: Thank you for your valuable comments. We have added further research at the end of the conclusion (Line 395–399).

References:

  1. Verify the formatting of your references.

Response: Thank you for your valuable suggestions. We have checked the formatting of references through the full text.

Comments on the Quality of English Language

  1. Moderate editing of the English language is required.

Response: Thank you for your valuable suggestions. The revised manuscript has been checked by XR Language Editing Services. Thank you for your time and patience.

Reviewer 3 Report

Comments and Suggestions for Authors

This manuscript describes the effect of feeding Galacto-Oligosaccharides on Holstein heifer calves. In the previous experiment, authors clarified the effect of feeding 10g of Galacto-Oligosaccharides. The manuscript ID animals-2625743 examines whether lower doses are effective and contains new information in the nutritional management of dairy calves. The materials and methods for this experiment are fine. The conclusions are consistent with the evidence and arguments presented. I thought this manuscript was highly complete. Please consider the following minor modifications.

L238-239 (Already abbreviated with L144-146) Average daily gain (ADG) and average daily feed intake (ADFI) => The ADG and ADFI

L252 (Already abbreviated with L88) short-chain fatty acids (SCFAs) => SCFAs

Author Response

Comments and Suggestions for Authors:

This manuscript describes the effect of feeding Galacto-Oligosaccharides on Holstein heifer calves. In the previous experiment, authors clarified the effect of feeding 10g of Galacto-Oligosaccharides. The manuscript ID animals-2625743 examines whether lower doses are effective and contains new information in the nutritional management of dairy calves. The materials and methods for this experiment are fine. The conclusions are consistent with the evidence and arguments presented. I thought this manuscript was highly complete. Please consider the following minor modifications.

  1. L238-239 (Already abbreviated with L144-146) Average daily gain (ADG) and average daily feed intake (ADFI) => The ADG and ADFI

Response: Thank you for your reminder. We have revised the abbreviations through the full text.

  1. L252 (Already abbreviated with L88) short-chain fatty acids (SCFAs) => SCFAs

Response: Thank you for your reminder, and we have revised the abbreviations through the full text.

Reviewer 4 Report

Comments and Suggestions for Authors

Overall, this paper adds to the large knowledge base of data regarding the use of Oligosaccharides for livestock production. This paper builds upon previous studies where 10g / head daily were utilized, by evaluating different GOS levels. Generally, there are some really good features of this paper though there are a couple of items which I have questions about.

General questions:

- Was the colostrum given to each calf the same nutritionally and based upon immunoglobulin content? Was this measured?

- What company provided the "normal milk" and did they also produce the colostrum?

- Was their any variation in the milk provided which could have influenced the mineral content of blood or was it consistent across treatments and across the trial period?

- Why were blood samples only collected at day 29? Would it not have been beneficial to take blood samples within the first 2 days of birth?

Specific questions:

- On line 118, is that immediately "after birth"?

- On line 121, should it be milk / colostrum?

- On line 130, is the starter which was collected only refusals or what type of samples?

- On line 133 & 134, change to "following the methods of" instead of "followed"

Comments on the Quality of English Language

Overall very good, just a few minor changes I would make. The biggest concern is multiple sentences start with abbreviations such as GOS which is generally not a good idea.

Author Response

Reviewer: 4

Comments and Suggestions for Authors:

Overall, this paper adds to the large knowledge base of data regarding the use of Oligosaccharides for livestock production. This paper builds upon previous studies where 10g / head daily were utilized, by evaluating different GOS levels. Generally, there are some really good features of this paper though there are a couple of items which I have questions about.

General questions:

  1. Was the colostrum given to each calf the same nutritionally and based upon immunoglobulin content? Was this measured?

Response: Thank you for your comments. The colostrum was nutritionally consistent and met the standards for colostrum management. The immunoglobulin G content in the colostrum was consistently greater than 50 g/L. Additionally, colostrum produced by the same batch of healthy cows was mixed in colostrum milk tank, then fed to newborn calves to eliminate the interindividual error.

  1. What company provided the "normal milk" and did they also produce the colostrum?

Response: Thank you for your reminder. To prevent barriers to understanding, we have changed "normal milk" into "heated raw milk". "heated raw milk" is raw milk sourced from the farm first heated to 42℃, and then naturally cooled to 38℃ before fed to calves. Both the heated raw milk and colostrum were produced in the dairy farm.

  1. Was their any variation in the milk provided which could have influenced the mineral content of blood or was it consistent across treatments and across the trial period?

Response: Thank you for your valuable comments. Both the colostrum and heated raw milk were mixed in milk tank before feeding to all calves. So the composition of milk which across different treatment groups and throughout the experimental period remained consistent. The milk underwent thorough analysis and quality control to ensure compliance with calf-rearing management standards. Therefore, we believe that there was no variation in this regard, eliminating the possibility of milk composition changes affecting blood mineral content.

  1. Why were blood samples only collected at day 29? Would it not have been beneficial to take blood samples within the first 2 days of birth?

Response: Thank you for your valuable comments. Firstly, considering that blood collection in the first two days after birth causes great stress to the calves, and secondly, we aimed to assess the cumulative effects of the experimental treatments over the entire experimental period. So we collect blood samples on the day immediately following the completion of the 28-day experimental period, which is day 29.

Specific questions:

  1. On line 118, is that immediately "after birth"?

Response: Thank you for your valuable comments. We transfer the calves to the clean calf hutch immediately after birth to prevent cross-infection

  1. On line 121, should it be milk / colostrum?

Response: Thank you for your valuable comments. The change has been made, as suggested.

  1. On line 130, is the starter which was collected only refusals or what type of samples?

Response: Thank you for your valuable comments. Through the experiment, calves were fed with the same starter and it was collected once a week before the morning feeding. All of the samples were mixed for nutrient levels analysis.

  1. On line 133 & 134, change to "following the methods of" instead of "followed"

Response: Thank you for your reminder, and we have revised it as suggested. (Line 151)

Comments on the Quality of English Language

  1. Overall very good, just a few minor changes I would make. The biggest concern is multiple sentences start with abbreviations such as GOS which is generally not a good idea.

Response: Thank you for your valuable suggestions. The revised manuscript has been checked by XR Language Editing Services. Thank you for your time and patience.

Round 2

Reviewer 1 Report

Comments and Suggestions for Authors

The authors have diligently addressed the review comments, significantly enhancing the paper's quality. As a result, it is now well-suited for publication.

Author Response

Comments and Suggestions for Authors:

The authors have diligently addressed the review comments, significantly enhancing the paper's quality. As a result, it is now well-suited for publication.

Response: We think highly of your valuable comments and suggestions. Thank you for your time and patience.

Reviewer 2 Report

Comments and Suggestions for Authors

The updated version was much appreciated, and it appears to be an improvement. 

More details see attachment.

Author Response

Comments and Suggestions for Authors:

The updated version was much appreciated, and it appears to be an improvement. More details see attachment.

Response: We think highly of your valuable comments and suggestions. We have revised our manuscript as you suggested. Thank you for your time and patience.

  1. Suggest to remove "treatments" form title due to word "supplementation" is sufficient.

Recommenced to heifer and remain only calves.

Response: Thank you for your valuable suggestions. We have revised the title as you suggested.

  1. Suggest the authors revised and write as a simple summary. Present version seem to be the authors copy and place from the text which contain objective, methodological and conclusion.

Response: Thank you for your valuable comments. We have rewritten the simple summary as you suggested.

  1. Indicate age..

Response: Thank you for your valuable comments. We have changed "similar age" to "1 day of age" to prevent barriers to understanding.

  1. This section seem like the authors discuss the result which is not suitable when inclusion were made. Abstract section should be not contain discussion. Some recommendation should place at the end of conclusion section.

Response: Thank you for your valuable suggestions. We have removed this part of the abstract in the revised manuscript.

  1. Are this objective were tested? If no, recommend to remove. This seem like recontamination and may place at the conclusion section.

Response: Thank you for your valuable suggestions. We have removed this part of the introduction in the revised manuscript.

  1. How old? Please detail.

Response: Thank you for your valuable comments. We have changed "similar age" to "1 day of age" to prevent barriers to understanding.

  1. P=0.05, is it significance? If so, superscript should be indicated.

Response: Thank you for your valuable comments. The results are shown with two significant figures. The specific number of the P-value is above 0.05. Thus the statistical significance is not labelled.
